# Enhancing Breast Cancer Detection through Advanced AI-Driven Ultrasound Technology: A Comprehensive Evaluation of Vis-BUS

**DOI:** 10.3390/diagnostics14171867

**Published:** 2024-08-26

**Authors:** Hyuksool Kwon, Seok Hwan Oh, Myeong-Gee Kim, Youngmin Kim, Guil Jung, Hyeon-Jik Lee, Sang-Yun Kim, Hyeon-Min Bae

**Affiliations:** 1Laboratory of Quantitative Ultrasound Imaging, Seoul National University Bundang Hospital, Seongnam 13620, Republic of Korea; jinuking3g@snubh.org (H.K.); joseph9337@kaist.ac.kr (S.H.O.); mgkim@barreleye.co.kr (M.-G.K.); 2Imaging Division, Department of Emergency Medicine, Seoul National University Bundang Hospital, Seongnam 13620, Republic of Korea; 3Barreleye Inc., 312, Teheran-ro, Gangnam-gu, Seoul 06221, Republic of Korea; 4Electrical Engineering Department, Korea Advanced Institute of Science and Technology, Daejeon 34141, Republic of Korea; youngmin2007@kaist.ac.kr (Y.K.); jgl97123@kaist.ac.kr (G.J.); dlguswlr0811@kaist.ac.kr (H.-J.L.); kmjmksy@kaist.ac.kr (S.-Y.K.)

**Keywords:** breast cancer, artificial intelligence, ultrasound, attenuation coefficient, diagnostic performance

## Abstract

This study aims to enhance breast cancer detection accuracy through an AI-driven ultrasound tool, Vis-BUS, developed by Barreleye Inc., Seoul, South Korea. Vis-BUS incorporates Lesion Detection AI (LD-AI) and Lesion Analysis AI (LA-AI), along with a Cancer Probability Score (CPS), to differentiate between benign and malignant breast lesions. A retrospective analysis was conducted on 258 breast ultrasound examinations to evaluate Vis-BUS’s performance. The primary methods included the application of LD-AI and LA-AI to b-mode ultrasound images and the generation of CPS for each lesion. Diagnostic accuracy was assessed using metrics such as the Area Under the Receiver Operating Characteristic curve (AUROC) and the Area Under the Precision-Recall curve (AUPRC). The study found that Vis-BUS achieved high diagnostic accuracy, with an AUROC of 0.964 and an AUPRC of 0.967, indicating its effectiveness in distinguishing between benign and malignant lesions. Logistic regression analysis identified that ‘Fatty’ lesion density had an extremely high odds ratio (OR) of 27.7781, suggesting potential convergence issues. The ‘Unknown’ density category had an OR of 0.3185, indicating a lower likelihood of correct classification. Medium and large lesion sizes were associated with lower likelihoods of correct classification, with ORs of 0.7891 and 0.8014, respectively. The presence of microcalcifications showed an OR of 1.360. Among Breast Imaging-Reporting and Data System categories, category C5 had a significantly higher OR of 10.173, reflecting a higher likelihood of correct classification. Vis-BUS significantly improves diagnostic precision and supports clinical decision-making in breast cancer screening. However, further refinement is needed in areas like lesion density characterization and calcification detection to optimize its performance.

## 1. Introduction

Breast cancer remains the most commonly diagnosed malignancy and the second leading cause of cancer-related mortality among women globally, with its incidence and mortality rates continually increasing [1]. Early detection and timely treatment significantly reduce mortality rates, thereby prompting extensive efforts to enhance the accuracy of early breast cancer detection. Various imaging modalities, such as mammography, breast ultrasound, and magnetic resonance imaging (MRI), are utilized for breast cancer screening and supplementary diagnostic purposes. Among these, breast ultrasound is frequently employed due to its safety, convenience, non-invasive nature, and high diagnostic accuracy [2]. Additionally, breast ultrasound proves advantageous for patients with dense breast tissue, where mammography often exhibits reduced sensitivity.

Despite its benefits, breast ultrasound is limited by issues such as lack of standardization, low reproducibility, high operator dependency, and lengthy examination times [3]. Recently, artificial intelligence (AI) has made significant strides in fields such as automatic speech recognition, image recognition, and natural language processing. In the realm of medical imaging, deep learning (DL) technologies have advanced notably, particularly in breast imaging [4]. AI enhances the utility of breast ultrasound by automating the recognition and quantitative assessment of imaging data [5]. Integrating AI into clinical breast ultrasound practices is crucial as it saves time, reduces physician fatigue, and compensates for potential deficits in experience and skill among healthcare providers [6].

In the field of AI-powered breast imaging, significant contributions include the development of software utilizing over 2 million proprietary ultrasound images and analyzing over 17,000 features per image. This approach has enhanced diagnostic accuracy and reduced false positives and benign biopsies [7]. A study published demonstrated that such software significantly improved diagnostic performance, particularly by reducing inter- and intra-observer variability in breast lesion assessments [8]. Another study highlighted the software’s perfect accuracy in diagnosing invasive lobular carcinoma, a particularly challenging type of breast cancer to detect [9]. These advancements underscore the transformative potential of AI in improving breast cancer screening and diagnosis. However, there are limitations to real-time identification and display of lesions while also determining their malignancy.

Vis-BUS (Barreleye Inc., Seoul, Republic of Korea), an AI-driven ultrasound solution, represents a significant leap forward in breast cancer detection technology (Appendix A). This innovative tool incorporates two main components: Lesion Detection AI (LD-AI) and Lesion Analysis AI (LA-AI). In addition to these components, Vis-BUS calculates and provides a Cancer Probability Score (CPS), which quantifies the likelihood of a lesion being malignant. The system also provides detailed information regarding Breast Imaging-Reporting and Data System (BI-RADS) categories, assisting clinicians in making informed decisions.

This study hypothesizes that the AI-driven ultrasound solution Vis-BUS significantly enhances diagnostic accuracy and efficiency in differentiating between benign and malignant breast lesions using CPS based on LD-AI and LA-AI compared to traditional methods. The primary objective is to evaluate Vis-BUS’s performance using CPS and its correlation with diagnostic accuracy, while also analyzing factors contributing to diagnostic discrepancies, including density of breast tissue, lesion size, calcification presence, and BI-RADS categories. This study aims to provide comprehensive insights into the capabilities, limitations, and clinical impact of AI-driven ultrasound solutions, guiding future developments and implementations in clinical settings to improve breast cancer detection accuracy and efficiency.

## 2. Materials and Methods

### 2.1. Background and Theoretical Framework

#### 2.1.1. Breast Imaging-Reporting and Data System (BI-RADS)

The Breast Imaging-Reporting and Data System (BI-RADS) is a standardized system developed by the American College of Radiology (ACR) for categorizing breast imaging findings, primarily from mammography, ultrasound, and MRI. BI-RADS categories range from 0 to 6, with each category indicating the level of suspicion for malignancy. For instance:BI-RADS 0: Incomplete; additional imaging evaluation needed.BI-RADS 1: Negative; no findings of concern.BI-RADS 2: Benign findings.BI-RADS 3: Probably benign; <2% risk of malignancy, with a recommendation for short-term follow-up.BI-RADS 4: Suspicious abnormality; subdivided into 4A, 4B, and 4C, with increasing likelihood of malignancy (2–95%).BI-RADS 5: Highly suggestive of malignancy; >95% risk, with biopsy recommended.BI-RADS 6: Known biopsy-proven malignancy.

Understanding these categories is essential as they guide clinical decision-making and patient management, influencing the need for further diagnostic procedures or immediate treatment.

#### 2.1.2. Artificial Intelligence in Medical Imaging

Artificial Intelligence (AI), particularly through deep learning techniques, has revolutionized the field of medical imaging by enhancing the accuracy and efficiency of image analysis. In breast cancer detection, AI-driven tools analyze large datasets of imaging data to identify patterns and features that may indicate the presence of malignancy. These tools assist radiologists by providing quantitative assessments, reducing variability in interpretation, and potentially lowering the rates of false positives and false negatives.

AI algorithms, such as convolutional neural networks (CNNs), are commonly used to analyze ultrasound images by identifying and classifying lesions. The AI tool evaluated in this study, Vis-BUS, incorporates advanced algorithms for lesion detection and analysis, as well as a Cancer Probability Score (CPS) to assess the likelihood of malignancy.

#### 2.1.3. Diagnostic Performance Metrics

In evaluating the effectiveness of diagnostic tools, several performance metrics are commonly used, including:Area Under the Receiver Operating Characteristic Curve (AUROC): A measure of the ability of a classifier to distinguish between classes. The AUROC value ranges from 0.5 (no better than random guessing) to 1.0 (perfect classification).Area Under the Precision–Recall Curve (AUPRC): Focuses on the trade-off between precision (positive predictive value) and recall (sensitivity), especially important in cases where one class is much less frequent than the other.

These metrics were used in our study to assess the diagnostic accuracy of the Vis-BUS AI tool, ensuring a comprehensive evaluation of its performance.

#### 2.1.4. Cancer Probability Score (CPS)

CPS is a quantitative measure generated by the Vis-BUS AI tool to assess the likelihood that a breast lesion is malignant. The CPS is calculated by analyzing various features of the lesion identified in ultrasound images, such as size, shape, margin characteristics, echo patterns, and the presence of microcalcifications. The AI algorithm processes these inputs using a fully convolutional neural network (FCN) that produces a probabilistic score ranging from −100 to 100, where higher scores indicate a higher likelihood of malignancy.

Calculation Process:Input Data: Ultrasound images of breast lesions are processed by the AI system.Feature Extraction: The AI system extracts and analyzes key features of the lesions, including morphological and textural characteristics.Score Generation: The AI uses the extracted features to generate the CPS, which reflects the probability of the lesion being malignant. The score is derived through a combination of lesion detection (using Lesion Detection AI) and lesion analysis (using Lesion Analysis AI).Output: The CPS is presented on a scale from −100 to 100, aiding clinicians in assessing the malignancy risk of the lesion.The CPS formula
$$CPS={{\theta_{LA-AI}(B}_{img},\theta }_{LD-AI}(B_{img}))$$where, $\theta_{LA-AI}$ and $\theta_{LD-AI}$ denotes LD-AI and LA-AI neural network, respectively.

#### 2.1.5. CPS and Its Relationship with AUPRC and AUROC

AUROC (Area Under the Receiver Operating Characteristic Curve): The AUROC measures the ability of the CPS to discriminate between benign and malignant lesions across various threshold settings. It is calculated by plotting the true positive rate (sensitivity) against the false positive rate (1-specificity) at different thresholds. An AUROC of 1 indicates perfect discrimination, while an AUROC of 0.5 indicates no better performance than random guessing.AUPRC (Area Under the Precision–Recall Curve): The AUPRC focuses on the trade-off between precision (positive predictive value) and recall (sensitivity), particularly in datasets where the classes (malignant vs. benign) are imbalanced. The AUPRC is calculated by plotting precision against recall at various threshold levels. A higher AUPRC indicates that the CPS maintains good precision and recall, even when the threshold is varied.

Numerical Relationship: The CPS serves as the primary input for calculating both AUROC and AUPRC. These metrics evaluate the performance of the CPS across different thresholds, providing a comprehensive assessment of its diagnostic accuracy. A high CPS correlates with high AUROC and AUPRC values, indicating that the score is effective in correctly classifying lesions as malignant or benign across a range of scenarios. Essentially, the AUROC and AUPRC provide a numerical summary of how well the CPS performs in distinguishing between different outcomes, with high values reflecting strong performance.

### 2.2. Study Design

This retrospective study aims to evaluate the diagnostic performance of Vis-BUS, an AI-driven ultrasound tool designed to distinguish between benign and malignant breast lesions. The study seeks to validate the effectiveness of Vis-BUS by focusing on its ability to improve accuracy in breast cancer screening and identifying critical factors affecting its performance (Figure 1).

### 2.3. Development of Vis-BUS

Figure 2 illustrates the overall configuration of the Vis-BUS neural network. Vis-BUS integrates two primary components: Lesion Detection AI (LD-AI) and Lesion Analysis AI (LA-AI). LD-AI uses b-mode ultrasound images, $B_{img}\sim R^{256\times 256}$, to identify the location of the lesion, while LA-AI analyzes the breast b-mode image characteristics. 

The development of Vis-BUS, an AI-driven ultrasound tool, was a multi-step process involving the integration of advanced machine learning techniques, particularly deep learning, to enhance the accuracy and efficiency of breast lesion detection and classification. 

#### 2.3.1. System Architecture and Components

Vis-BUS consists of two primary AI components:Lesion Detection AI (LD-AI): This module is responsible for detecting the location of the breast lesions in ultrasound images.Lesion Analysis AI (LA-AI): This module further analyzes the detected lesions to assess their malignancy and generates the Cancer Probability Score (CPS).

LD-AI (Lesion Detection AI):The LD-AI module was developed based on a state-of-the-art object detection framework, leveraging deep convolutional neural networks (CNNs) to identify lesions within B-mode ultrasound images. The LD-AI component includes the following key elements:The LD-AI is implemented based on the object detection framework [10], where the neural network outputs lesion position and size and a corresponding confidence score as an output ($L_{coord})$. The LD-AI consists of a convolutional encoder backbone and a feature network. The encoder backbone is based on the EfficientNet [11] for parameter-efficient feature analysis of the $B_{img}$. A bi-directional feature network [10] is employed as the feature network. The lesion location is trained to minimize mean squared error loss between the ground truth and the LD-AI output, while the confidence score employs focal loss [12] as a learning objective. The AdamW [13] with a learning rate of 1 × 10^−4^ is utilized as the network optimizer.Convolutional Encoder Backbone: The backbone of LD-AI is based on EfficientNet, a highly efficient CNN architecture known for its balance between performance and computational cost. EfficientNet was chosen for its ability to effectively capture the complex features of ultrasound images with fewer parameters.Feature Network: A bi-directional feature network is employed, allowing the system to analyze features from multiple scales, improving the detection of lesions of varying sizes and shapes.Training Process: The LD-AI was trained using a large dataset of 19,000 annotated ultrasound images. The ground truth for lesion locations was provided by expert radiologists. The network was optimized with a learning rate set to 1 × 10^−4^, focusing on minimizing the mean squared error between the predicted and actual lesion coordinates.LA-AI (Lesion Analysis AI):Following the detection of a lesion, the LA-AI module processes the identified lesion to determine its malignancy, outputting a CPS. The LA-AI component includes:The LA-AI employs $B_{img}$, $L_{coord}$, and lesion image to analyze the lesion malignancy. Features of each input are interpreted using a fully convolutional neural network (FCN encoder). These features are then concatenated channel-wise into a latent vector, which serves as the input for the fusion encoder, producing CPS as the neural network output. The fusion encoder architecture is based on EfficientNet. The LA-AI is optimized with the AdamW optimizer (learning rate = 1 × 10^−3^), with a learning objective of minimizing binary cross entropy between the CPS and the ground truth. Fully Convolutional Network (FCN) Encoder: The FCN processes the B-mode images and the coordinates of the detected lesions. This encoder extracts features that are crucial for malignancy assessment.Fusion Encoder: The output from the FCN is concatenated channel-wise into a latent vector, which is then passed through the fusion encoder. This architecture, also based on EfficientNet, integrates the features extracted from the lesion to generate a CPS.Optimization: The LA-AI was trained at a learning rate of 1 × 10^−3^, with the objective of minimizing binary cross-entropy between the CPS and the actual malignancy status as determined by biopsy.

#### 2.3.2. Training and Data Preparation

Data Sources:The training dataset consisted of ultrasound images acquired from machines by leading manufacturers such as Philips (Amsterdam, The Netherlands), GE (Boston, MA, USA), and Fujifilm (Tokyo, Japan). This diversity in data sources was crucial to ensure that Vis-BUS could perform reliably across different imaging systems.Annotation and Ground Truth:Each ultrasound image was meticulously annotated by expert radiologists at Seoul National University Bundang Hospital (SNUBH), providing the ground truth for lesion location and malignancy. This rigorous annotation process was critical for training the AI models to achieve high accuracy in real-world clinical settings.Model Validation and Testing:To ensure the robustness of Vis-BUS, the models were validated on a separate set of ultrasound images not used in training. The performance of the system was evaluated using key metrics such as the Area Under the Receiver Operating Characteristic Curve (AUROC) and the Area Under the Precision–Recall Curve (AUPRC).

#### 2.3.3. Implementation and Integration

Vis-BUS operates by connecting to the HDMI port of conventional ultrasound imaging devices, allowing real-time processing of ultrasound video feeds. The software is installed on a tablet, which serves as the interface for clinicians. This setup enables:Real-Time Analysis: The AI processes the video feed in real-time, providing immediate diagnostic insights as the ultrasound examination is being conducted.Freeze-Frame Analysis: Clinicians can pause the ultrasound scan and perform detailed analysis on static images using the AI tool, which continues to offer diagnostic information without losing any capabilities.User Interface: The tablet interface is designed to be intuitive, allowing clinicians to easily navigate through diagnostic results, adjust settings, and generate detailed reports.

#### 2.3.4. Cancer Probability Score (CPS)

The CPS, generated by the LA-AI module, quantifies the likelihood of a lesion being malignant. This score is displayed on a scale from −100 to 100, with higher scores indicating a greater probability of malignancy. The CPS is a crucial component that aids clinicians in making informed decisions about patient management.

#### 2.3.5. Performance and Optimization

The development process included continuous iterations of training and validation to optimize the performance of Vis-BUS. The system’s high AUROC and AUPRC values reflect the efficacy of the training process and the robustness of the AI algorithms in accurately diagnosing breast lesions.

For the training of LD-AI and LA-AI, 19k breast ultrasound images are acquired using ultrasound machines from Philips (NL), GE (US), and Fujifilm (JP). The ground truth lesion location and lesion malignancy are annotated by an expert radiologist in SNUBH under IRB approval (IRB number: B-2301-807-108).

### 2.4. Explanation of Vis-BUS Operation

Vis-BUS operates by connecting a tablet, equipped with the Vis-BUS software, to the HDMI port of an existing ultrasound imaging device. This connection allows the tablet to receive the ultrasound video feed directly. Once connected, the Vis-BUS software begins to display the live video feed from the ultrasound device and can analyze the ultrasound images in real-time, providing immediate feedback and diagnostic information as the images are being captured.

Additionally, the Vis-BUS software supports analysis on freeze frames. When the ultrasound image is paused or frozen, the software can still perform detailed analysis on the static image. This feature allows for thorough examination and evaluation of specific frames without losing any diagnostic capabilities. During continuous ultrasound scanning, Vis-BUS provides ongoing analysis, highlighting areas of interest, detecting anomalies, and offering real-time diagnostic insights.

The tablet interface is designed to be intuitive and user-friendly, allowing clinicians to easily navigate through the analysis results, adjust settings, and interact with the diagnostic tools provided by the Vis-BUS software. The software can generate detailed diagnostic reports based on both real-time and frozen images, which can be saved, shared, or integrated into the patient’s medical record.

Vis-BUS also provides quantitative information on the likelihood of a lesion being benign or malignant, displaying this information on a scale from −100 to 100 on the screen. Additionally, it offers detailed BI-RADS values, assisting clinicians in making more informed diagnostic decisions.

### 2.5. Patients Dataset

The dataset for this study consisted of 258 breast ultrasound examinations Seoul National University Bundang Hospital. All data were used following Institutional Review Board approval (IRB number: B-2301-807-108), anonymized, and strictly prohibited from being shared externally. Each examination was categorized by pathologic results of the biopsy as either benign (*n* = 129) or malignant (*n* = 129). The dataset included a diverse range of breast lesion characteristics to ensure comprehensive evaluation of the AI tool’s performance across various clinical scenarios. These data were stored in the Vis-BUS analysis data folder, where each image was automatically loaded and analyzed.

### 2.6. Evaluation Metrics and Statistical Analysis

The diagnostic performance of Vis-BUS was primarily evaluated using the following metrics. Area under the receiver operating characteristic (AUROC) is a performance measurement for classification problems at various threshold settings. It represents the probability that a randomly chosen positive instance is correctly ranked with greater suspicion than a randomly chosen negative instance. An AUROC value ranges from 0 to 1, with 1 indicating perfect accuracy and 0.5 indicating performance no better than random guessing.

Area under the precision–recall curve (AUPRC) focuses on the performance of a classifier in terms of precision (positive predictive value) and recall (sensitivity). The AUPRC provides a single value that summarizes the trade-off between these two metrics across different thresholds. High AUPRC values indicate that the classifier maintains high precision and recall even at varying threshold levels.

The overall diagnostic accuracy was assessed by computing these metrics for the entire dataset, providing a comprehensive evaluation of Vis-BUS’s ability to distinguish between benign and malignant lesions. The gold standard for determining the true nature of the breast lesions was the biopsy results, which provided definitive pathological diagnoses.

To compare the characteristics of benign and malignant breast masses, we used *t*-tests and chi-square tests. The *t*-test compared the means of numerical variables, while the chi-square test assessed associations between categorical variables and the ground truth. A *p*-value < 0.05 indicates a significant difference or association.

The study evaluated the impacts of lesion density and lesion size on diagnostic accuracy, positing that lesions with lower density and smaller size are more susceptible to misclassification. Additionally, the presence or absence of calcifications was analyzed to ascertain its effect on the diagnostic performance of Vis-BUS. The influence of various BI-RADS categories on diagnostic accuracy was also examined, with particular attention given to lower BI-RADS categories, which are hypothesized to be more challenging to classify accurately. Logistic regression analysis was employed to investigate the relationship between the likelihood of misclassification and several factors, including lesion density, lesion size, presence of calcifications, and BI-RADS categories. The analysis yielded Odds Ratios (OR) with 95% confidence intervals (CI) for each factor, providing insights into the strength and direction of their association with diagnostic errors.

## 3. Results

### 3.1. Vis-BUS Operational Outcomes

The implementation of Vis-BUS has shown that it effectively processes both live and static ultrasound images when connected to the HDMI port of conventional ultrasound equipment. Vis-BUS provides real-time diagnostic insights and detailed post-analysis on freeze frames.

In clinical tests with various ultrasound machines, Vis-BUS consistently delivered high-resolution, quantitative analyses. The software accurately identified key pathological features such as tissue texture, echo enhancement, and artifacts. It also maintained diagnostic precision when analyzing freeze frames, allowing for detailed examination of specific frames. Table 1 provides a comprehensive summary of the experimentations and outcomes highlighting the key features of the Vis-BUS AI tool, including its real-time diagnostic capabilities, performance on freeze-frame analysis, effectiveness in lesion characterization, and consistency across various ultrasound machines.

Vis-BUS provided quantitative information on the likelihood of a lesion being benign or malignant, displayed on a scale from −100 to 100, along with detailed BI-RADS values. This feature assisted clinicians in making more informed diagnostic decisions (Figure 3).

### 3.2. Patient Characteristics

The patient characteristics is provided in Table 2. Patients with malignant masses are generally older (average age 51.57 years) compared to those with benign masses (average age 44.91 years), with a significant *p*-value of <0.001. Malignant lesions are larger, averaging 2.41 cm compared to 1.02 cm for benign lesions, again with a *p*-value of <0.001. T-stage distribution differs significantly, with most benign cases being T1 (123 cases) and most malignant cases spread across T1 (70 cases), T2 (48 cases), and T3 (11 cases), reflecting more advanced tumors in malignancies (*p*-value < 0.001). In terms of breast density, 116 benign cases are classified as ‘Dense’ compared to 104 malignant cases, with malignant cases also having a notable proportion of ‘Unknown’ density (22 cases), indicating diagnostic challenges (*p*-value < 0.001). Microcalcifications are more common in malignant cases (66 cases) than benign ones (13 cases), with a *p*-value of <0.001. The shape of lesions also varies significantly; malignant lesions are predominantly irregular (124 cases), while benign lesions are more often oval (58 cases) or irregular (66 cases) (*p*-value < 0.001). Additionally, the margins of malignant lesions are more likely to be indistinct, spiculated, or angular, whereas benign lesions tend to be indistinct or circumscribed (*p*-value < 0.001). Lastly, the echo-patterns differ, with malignant lesions being more likely to have heterogeneous and complex patterns compared to benign ones, which are primarily hypoechoic or isoechoic (*p*-value < 0.001). 

### 3.3. Cancer Probability Score (CPS) Performance in Diagnosing Breast Masses

To evaluate the effectiveness of the CPS in diagnosing breast masses as benign or malignant, we analyzed two key performance metrics: the area under the receiver operating characteristic curve (AUROC) and the area under the precision–recall curve (AUPRC) (Figure 4). The AUROC measures the ability of the CPS to correctly distinguish between benign and malignant breast masses. An AUROC value of 0.964 indicates a very high level of accuracy, meaning that the CPS is highly effective in differentiating between benign and malignant cases with performance close to perfect discrimination.

The AUPRC evaluates the precision (positive predictive value) and recall (sensitivity) of the CPS, which is particularly important in datasets where one class may be less frequent (e.g., malignant cases). An AUPRC value of 0.967 suggests that the CPS maintains excellent precision and recall, effectively identifying malignant cases while minimizing false positives and false negatives. These high values indicate that the CPS is a reliable tool for accurately distinguishing between benign and malignant breast masses, making it a valuable asset in clinical practice for the diagnosis and management of breast cancer.

### 3.4. Logistic Regression Analysis of Diagnostic Accuracy

The logistic regression analysis aimed to identify factors contributing to diagnostic discrepancies, focusing on lesion density, lesion size, presence of calcifications, and BI-RADS categories (Table 3). The analysis revealed that the ‘Fatty’ lesion density category had an extremely high odds ratio (OR) of 27.7781, indicating potential convergence issues. This suggests that the model might not have accurately captured the effect of this category due to data limitations or inherent variability. On the other hand, the ‘UK’ category had an OR of 0.3185, significantly less than 1, suggesting it is associated with a lower likelihood of correct classification. This could imply that unknown density cases are more prone to diagnostic errors, potentially due to the ambiguity in their classification.

For lesion size, both medium and large sizes had ORs less than 1 (0.7891 and 0.8014, respectively), indicating a trend towards a lower likelihood of correct classification compared to smaller lesions. However, these results were not statistically significant, suggesting that while there might be a trend, the effect of lesion size on diagnostic accuracy is not strong enough to be definitive. The presence of microcalcifications showed an OR of 1.360, suggesting a higher likelihood of misclassification, but this result was also not statistically significant. Among the BI-RADS categories, category C5 had a significantly higher OR of 10.173, indicating a much higher likelihood of correct classification. This suggests that higher BI-RADS categories, which typically represent more suspicious findings, are more accurately diagnosed, reflecting the effectiveness of diagnostic processes for more evident cases.

## 4. Discussion

The findings from this study highlight the significant potential of the Vis-BUS AI-driven ultrasound system in enhancing breast cancer diagnostic accuracy. The system’s ability to analyze lesion characteristics and provide a Cancer Probability Score (CPS) has demonstrated considerable effectiveness, particularly in distinguishing between benign and malignant breast lesions. The logistic regression analysis provided deeper insights into specific factors affecting diagnostic accuracy, such as lesion density, size, presence of calcifications, and BI-RADS categories.

The transformative potential of AI in improving breast cancer screening and diagnosis has been highlighted in various studies. A systematic review explored the effectiveness of AI applications in breast cancer diagnosis, emphasizing their potential in clinical settings [9]. Another study discussed advanced machine learning techniques for segmenting breast ultrasound images, which can enhance lesion detection and characterization accuracy [14]. The integration of different advanced imaging modalities, including AI-enhanced ultrasound, has been shown to optimize patient outcomes and improve breast cancer diagnostic accuracy [15].

The results from our study demonstrate that the AI tool, Vis-BUS, performs exceptionally well in identifying lesions classified as BI-RADS 5. While this finding confirms the reliability of the tool, we acknowledge that BI-RADS 5 lesions, by definition, already carry a greater than 95% likelihood of malignancy and typically lead to biopsy and definitive treatment without much diagnostic ambiguity [16]. Therefore, the AI’s effectiveness in these cases may not offer significant clinical value, as these lesions are generally well-recognized as malignant by experienced radiologists.

The BI-RADS is widely used to standardize breast imaging findings, with categories ranging from 0 to 6, where BI-RADS 5 indicates a high suspicion of malignancy. The primary clinical challenge and, consequently, the potential for AI assistance, often lie in the intermediate categories, particularly BI-RADS 3 (probably benign) and BI-RADS 4 (suspicious abnormality). These categories encompass lesions where the probability of malignancy is less definitive, ranging from 2% to 95%, depending on sub-categorization. Accurate diagnosis in these categories is crucial for balancing the risks of unnecessary biopsies with the need for timely cancer detection [17,18].

Our findings suggest that the AI tool’s true value may be more pronounced in these lower BI-RADS categories, where diagnostic uncertainty is higher and where AI could significantly aid in decision-making by reducing inter-observer variability and potentially decreasing the number of unnecessary biopsies [2,19]. For example, studies have shown that AI algorithms can improve the diagnostic accuracy of radiologists, particularly in more ambiguous cases [20,21]. Moving forward, future studies should focus on evaluating the AI tool’s performance specifically in BI-RADS 3 and 4 lesions. By refining the AI’s algorithms to better assess these intermediate cases, we can better harness the tool’s potential to assist in more challenging diagnostic scenarios, thereby improving clinical outcomes and patient care.

Subgroup analysis of BI-RADS 3 and 4 categories is critical for understanding the clinical impact of AI tools like Vis-BUS, particularly in reducing false negative and false positive rates. For BI-RADS 3, minimizing false negatives is essential to avoid delayed detection of malignancies, which, though rare, can significantly affect patient outcomes [22]. Conversely, in BI-RADS 4, reducing false positives is crucial to prevent unnecessary biopsies and the associated anxiety and costs. Examining cases where the AI either misses malignancies in BI-RADS 3 or incorrectly flags benign lesions in BI-RADS 4 provides valuable insights into the tool’s limitations and areas for improvement [23]. Future studies should focus on these analyses to refine AI algorithms and enhance their clinical utility, ensuring more accurate and consistent diagnostic outcomes.

The performance metrics for CPS, specifically the Area Under the Receiver Operating Characteristic curve (AUROC) and the Area Under the Precision–Recall curve (AUPRC), were evaluated. The AUROC of 0.964 indicates a very high level of accuracy, suggesting that the CPS is highly effective in differentiating between benign and malignant cases with performance close to perfect discrimination. The AUPRC value of 0.967 suggests that the CPS maintains excellent precision and recall, effectively identifying malignant cases while minimizing false positives and false negatives. These high values indicate that the CPS is a reliable tool for accurately distinguishing between benign and malignant breast masses, making it a valuable asset in clinical practice for the diagnosis and management of breast cancer.

Lesion density emerged as a notable factor, with the ‘Fatty’ category showing an extremely high odds ratio (OR), indicating potential convergence issues. This suggests that the model might not have accurately captured the effects of this category due to inherent variability or data limitations. Conversely, the ‘Unknown’ density category had a significantly lower OR, implying that these cases are more prone to diagnostic errors, possibly due to the ambiguity in their classification. This finding underscores the importance of accurate lesion density categorization in improving diagnostic outcomes.

Lesion size also showed a trend where medium and large lesions were associated with lower likelihoods of correct classification compared to smaller lesions, although these findings were not statistically significant. This suggests that while size plays a role, other factors might equally influence diagnostic accuracy. The presence of microcalcifications was associated with a higher likelihood of misclassification, though this result was also not statistically significant. Among the BI-RADS categories, category C5 significantly improved diagnostic accuracy, reflecting the system’s effectiveness in identifying more suspicious and potentially malignant lesions. This aligns with clinical expectations and highlights the AI system’s potential in supporting clinical decision-making.

These findings suggest that while AI-driven systems like Vis-BUS offer significant improvements in diagnostic accuracy, certain challenges remain. The variability in lesion density, the ambiguous nature of some classifications, and the need for accurate characterization of lesion size and calcifications highlight areas where further refinement is necessary. Additionally, the excellent performance metrics of CPS underscore the importance of integrating advanced AI technologies into clinical workflows to enhance diagnostic outcomes and support clinicians in making more informed decisions. Future research will systematically compare the diagnostic performance of Vis-BUS against clinicians with different levels of experience in a controlled, real-time setting. By doing so, we aim to provide a comprehensive evaluation of how the AI tool can complement and enhance clinical decision-making, particularly for those with less experience.

This study, being retrospective in nature, carries inherent limitations that must be acknowledged. First, the lack of prospective comparisons with clinicians of varying experience levels limits the generalizability of our findings to broader clinical settings. Additionally, the retrospective analysis relies on archived ultrasound images, which may not fully represent the variability encountered in real-time diagnostic scenarios. The dataset used, while extensive, was sourced from a specific clinical environment, potentially introducing bias and limiting the applicability of our results to other settings. Finally, the reliance on a single AI tool without comparison to alternative methods or tools means that the observed outcomes might not reflect the full spectrum of possibilities available in current clinical practice. These limitations underscore the need for future prospective studies that can address these concerns and provide more robust evidence for the effectiveness of AI-driven diagnostic tools like Vis-BUS.

## 5. Conclusions

In conclusion, the Vis-BUS AI-driven ultrasound system represents a significant advancement in breast cancer diagnostics. Its ability to provide real-time, quantitative analysis of ultrasound images enhances diagnostic accuracy, particularly in identifying malignant lesions. The performance metrics for CPS, including AUROC and AUPRC, demonstrated the system’s high accuracy and reliability in distinguishing between benign and malignant lesions.

## Figures and Tables

**Figure 1 diagnostics-14-01867-f001:**
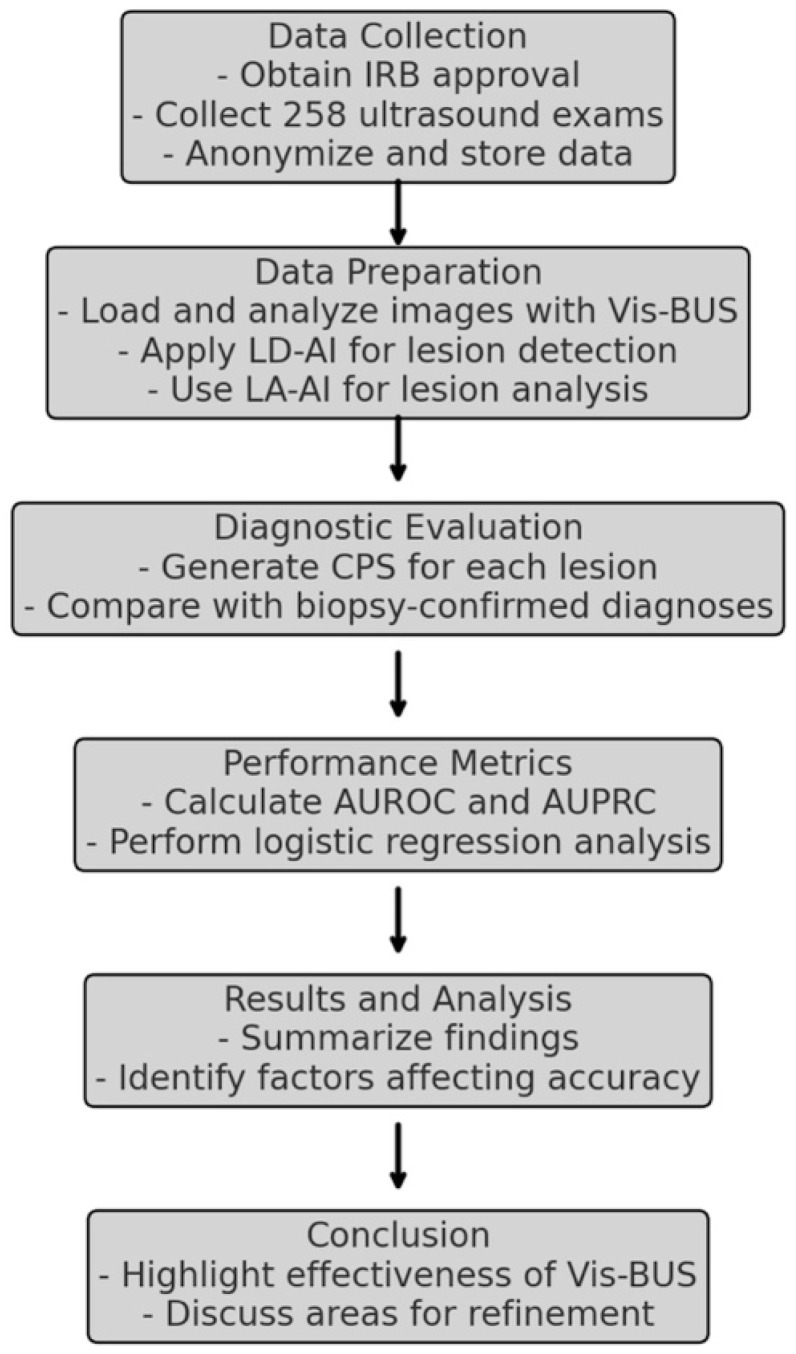
Study Flow for Evaluating Vis-BUS AI-Driven Ultrasound Tool.

**Figure 2 diagnostics-14-01867-f002:**
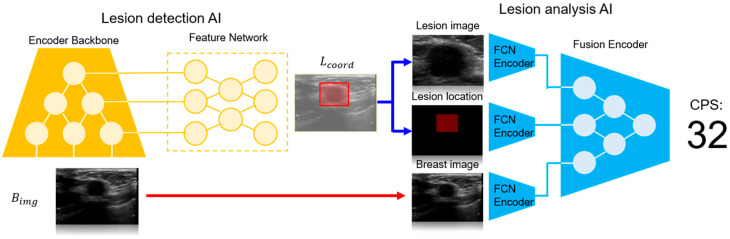
Overall configuration of the Vis-BUS neural networks.

**Figure 3 diagnostics-14-01867-f003:**
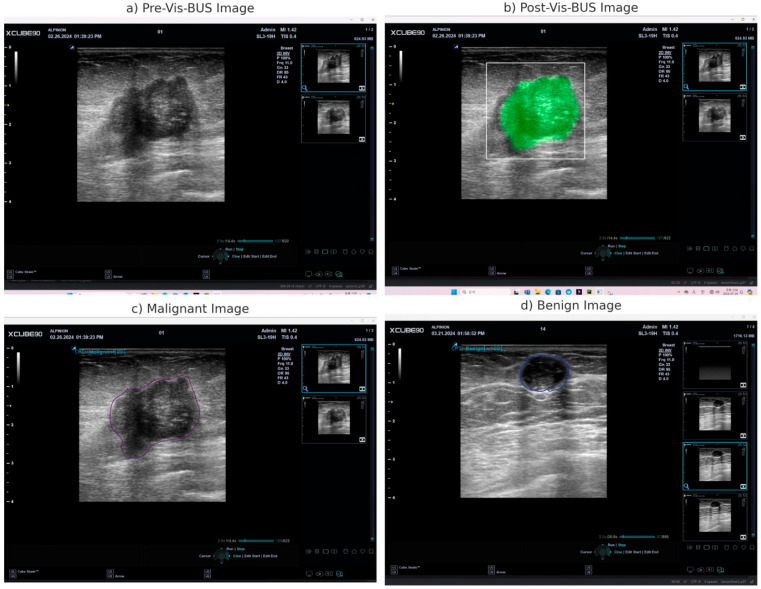
Ultrasound Images with Vis-BUS Analysis. (**a**) Pre-Vis-BUS Image: This image represents the ultrasound scan before applying the Vis-BUS AI-driven analysis. The lesion is visible but not yet processed by the AI tool. (**b**) Post-Vis-BUS Image: This image shows the same ultrasound scan after applying the Vis-BUS analysis. The lesion is highlighted in green, indicating the area identified by the AI for further evaluation. (**c**) Malignant Image: This image depicts a lesion identified by the Vis-BUS tool as malignant. The AI-generated outline and the Cancer Probability Score (CPS) are displayed, indicating a high likelihood of malignancy. (**d**) Benign Image: This image shows a lesion identified by the Vis-BUS tool as benign. The AI-generated outline and the CPS are displayed, indicating a low likelihood of malignancy.

**Figure 4 diagnostics-14-01867-f004:**
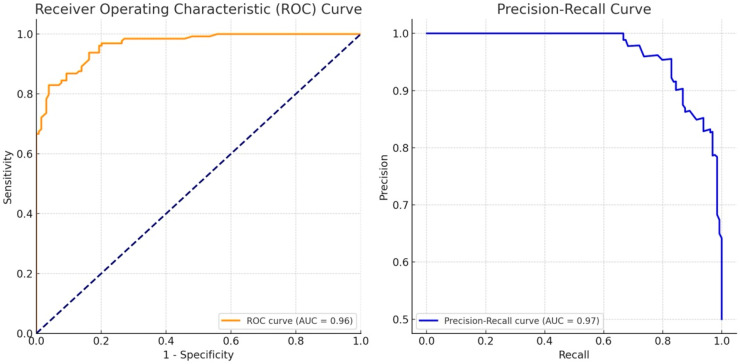
Cancer Probability Score Performance in Diagnosing Breast Masses.

**Table 1 diagnostics-14-01867-t001:** Summary of Experimentations and Outcomes.

Experiment/Outcome	Description	Key Findings
Real-time Diagnostic Insights	Vis-BUS processes live ultrasound images and provides immediate diagnostic feedback.	High-resolution quantitative analysis with real-time feedback, effectively highlighting pathological features.
Post-Analysis on Freeze Frames	Detailed examination of static ultrasound images using the AI tool.	Maintains diagnostic precision even when analyzing static images, ensuring thorough evaluation of specific frames.
Cancer Probability Score (CPS)	Quantitative assessment of the likelihood of a lesion being malignant or benign, displayed on a scale from −100 to 100.	Accurate differentiation between benign and malignant lesions, aiding in informed clinical decision-making.
Diagnostic Performance Across Ultrasound Machines	Evaluation of Vis-BUS performance on ultrasound images obtained from various machines.	Consistent performance across different ultrasound equipment, validating the tool’s versatility and reliability.
Lesion Characterization	Analysis of key pathological features such as tissue texture, echo patterns, and microcalcifications.	Precise identification of lesion characteristics, contributing to the accurate classification of breast lesions.

**Table 2 diagnostics-14-01867-t002:** Patients’ characteristics.

Variable	Benign (*n* = 129)		Malignancy (*n* = 129)		*p*-Value
age, mean (SD)	44.91	(9.99)	51.57	(10.9)	<0.001
lesion size, mean (SD)	1.02	(0.64)	2.41	(1.29)	<0.001
T-stage	T1	123	T1	70	<0.001
	T2	6	T2	48	
	T3	0	T3	11	
Density	Dense	116	Dense	104	<0.001
	Fatty	13	Fatty	3	
	Unknown	0	Unknown	22	
Microcalcification	Y	13	Y	66	<0.001
	N	116	N	63	
Ultrasound shape	Irregular	66	Irregular	124	<0.001
	Oval	58	Oval	4	
	Round	5	Round	1	
Ultrasound margin	Independent:	85	Independent:	91	<0.001
	Circumscribe	41	Circumscribe	2	
	Microlobulate	3	Microlobulate	3	
	Angular	0	Angular	8	
Ultrasound echo-pattern	Hypoechioic	96	Hypoechioic	114	<0.001
	Isoechoic	25	Isoechoic	3	
	Heterogeneous	9	Heterogeneous	9	
	Complex	0	Complex	3	

SD, standard deviation.

**Table 3 diagnostics-14-01867-t003:** Logistic Regression Analysis of Diagnostic Accuracy.

Variable	Odds Ratio (95% CI)	*p*-Value
Lesion Density		
Fatty	27.7781 (0.00, 1.00 × 10^6^)	1.0000
Unknown	0.3185 (0.123, 0.774)	0.0162
Lesion Size		
Medium	0.7891 (0.451, 1.376)	0.3980
Large	0.8014 (0.388, 1.658)	0.5530
Microcalcifications		
Present	1.360 (0.860, 2.160)	0.1941
BI-RADS Categories		
C3	2.548 (0.755, 8.602)	0.1318
C4a	0.7977 (0.289, 2.198)	0.6622
C4b	1.086 (0.242, 4.873)	0.9138
C4c	1.778 (0.186, 17.024)	0.6177
C5	10.173 (2.526, 40.962)	0.0011

## Data Availability

The data presented in this study are available on request from the corresponding author. The data are not publicly available due to Barreleye Inc. having intellectual property rights.

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
