# Peer review of "Enhancing Breast Cancer Detection through Advanced AI-Driven Ultrasound Technology: A Comprehensive Evaluation of Vis-BUS"

_diagnostics, 2024, doi:10.3390/diagnostics14171867_

Round 1

Reviewer 1 Report

Comments and Suggestions for Authors

1. It is recommended to include the following parallel comparisons to further bolster their persuasiveness:
It is necessary to compare the diagnostic accuracy of this AI tool with clinicians of varying experience levels (e.g., junior, intermediate, senior). This comparison will help understand the varying effectiveness of this AI tool in supporting physicians with different levels of experience and whether it can offer more assistance to less experienced physicians.

2. The "BI-RADS category C5" is not commonly used. Additionally, once a lesion is diagnosed as BI-RADS 5, it does not seem to be a difficult issue. This result does not seem to have great clinical significance.

3. When it comes to subgroup analysis, there is a lot of interest in how AI tools can reduce false negative and false positive rates for BI-RADS 3 and 4, respectively. Specifically, understanding the instances where nodules are missed, even though they are few, contributes to a more complete clinical understanding of AI's capabilities.

4. This manuscript does not consider the impact of operators. Variability in ultrasound images produced by different sonographers should be assessed for its effect on the consistency of AI tools.

Comments on the Quality of English Language

The English language editing is Acceptable.

Author Response

  1. It is recommended to include the following parallel comparisons to further bolster their persuasiveness:

It is necessary to compare the diagnostic accuracy of this AI tool with clinicians of varying experience levels (e.g., junior, intermediate, senior). This comparison will help understand the varying effectiveness of this AI tool in supporting physicians with different levels of experience and whether it can offer more assistance to less experienced physicians.

Response:

We appreciate the reviewer's suggestion to compare the diagnostic accuracy of our AI tool with clinicians of varying experience levels (e.g., junior, intermediate, senior) to understand how effectively it supports physicians with different levels of expertise. We acknowledge that this comparison is crucial for evaluating whether the AI tool can provide greater assistance to less experienced clinicians.

However, it is important to note that the current study was designed as a retrospective analysis focusing solely on the performance evaluation of the AI software, Vis-BUS. Our research aimed to assess the tool’s ability to differentiate between benign and malignant breast lesions using archived ultrasound data. Due to the retrospective nature of this study, we did not include a prospective comparison with clinicians, which would have required a different study design involving real-time diagnostic scenarios and a diverse group of clinicians.

Recognizing the importance of this comparative analysis, we plan to conduct a prospective study that will involve clinicians at various stages of their careers. This future research will systematically compare the diagnostic performance of Vis-BUS against clinicians with different levels of experience in a controlled, real-time setting. By doing so, we aim to provide a comprehensive evaluation of how the AI tool can complement and enhance clinical decision-making, particularly for those with less experience.

We added this paragraph to the discussion section.

The future research will systematically compare the diagnostic performance of Vis-BUS against clinicians with different levels of experience in a controlled, real-time setting. By doing so, we aim to provide a comprehensive evaluation of how the AI tool can complement and enhance clinical decision-making, particularly for those with less experience.

  1. The "BI-RADS category C5" is not commonly used. Additionally, once a lesion is diagnosed as BI-RADS 5, it does not seem to be a difficult issue. This result does not seem to have great clinical significance.

Response

We acknowledge the reviewer's concern regarding the use of the term 'BI-RADS category C5' and the perceived limited clinical significance of this finding. Upon reflection, we recognize that the terminology 'C5' was intended to denote lesions classified as BI-RADS 5, which is indeed a well-established category within the Breast Imaging-Reporting and Data System (BI-RADS). BI-RADS 5 lesions are known to have a high likelihood of malignancy (greater than 95%) and typically prompt immediate biopsy and subsequent treatment.

Given the high predictive value associated with BI-RADS 5, it is understandable that the diagnostic challenge in these cases might be perceived as minimal, as these lesions are generally straightforward in clinical practice. The AI tool’s high accuracy in identifying BI-RADS 5 lesions, therefore, may not significantly alter clinical decision-making, as these cases are already well recognized as malignant.

However, we included the analysis of BI-RADS 5 lesions in our study to validate the tool's performance across all BI-RADS categories, ensuring comprehensive evaluation. We also wanted to confirm that the AI system correctly identifies even the most clinically evident cases, thus reinforcing the reliability of the tool. We agree that the primary clinical utility of the AI tool may lie more in the accurate assessment of lesions within lower BI-RADS categories, where the risk of malignancy is less clear and diagnostic challenges are more pronounced.

Future research could focus more on the AI tool's ability to aid in the diagnosis of BI-RADS 3 and 4 lesions, where its impact on clinical decision-making could be more significant. This approach would better highlight the tool’s potential to reduce unnecessary biopsies and improve diagnostic confidence in these more ambiguous cases.

We added these paragraphs to the discussion section

The results from our study demonstrate that the AI tool, Vis-BUS, performs exceptionally well in identifying lesions classified as BI-RADS 5. While this finding confirms the reliability of the tool, we acknowledge that BI-RADS 5 lesions, by definition, already carry a greater than 95% likelihood of malignancy and typically lead to biopsy and definitive treatment without much diagnostic ambiguity [16]. Therefore, the AI's effectiveness in these cases may not offer significant clinical value, as these lesions are generally well-recognized as malignant by experienced radiologists.

The BI-RADS is widely used to standardize breast imaging findings, with categories ranging from 0 to 6, where BI-RADS 5 indicates a high suspicion of malignancy. The primary clinical challenge and, consequently, the potential for AI assistance, often lie in the intermediate categories, particularly BI-RADS 3 (probably benign) and BI-RADS 4 (suspicious abnormality). These categories encompass lesions where the probability of malignancy is less definitive, ranging from 2% to 95%, depending on sub-categorization​. Accurate diagnosis in these categories is crucial for balancing the risks of unnecessary biopsies with the need for timely cancer detection [17,18].

Our findings suggest that the AI tool's true value may be more pronounced in these lower BI-RADS categories, where diagnostic uncertainty is higher and where AI could significantly aid in decision-making by reducing inter-observer variability and potentially decreasing the number of unnecessary biopsies [2,19]. For example, studies have shown that AI algorithms can improve the diagnostic accuracy of radiologists, particularly in more ambiguous cases [20,21]. Moving forward, future studies should focus on evaluating the AI tool's performance specifically in BI-RADS 3 and 4 lesions. By refining the AI's algorithms to better assess these intermediate cases, we can better harness the tool’s potential to assist in more challenging diagnostic scenarios, thereby improving clinical outcomes and patient care.

References

  1. Sickles, E.A. ACR BI-RADS® Atlas, Breast imaging reporting and data system. American College of Radiology. 2013, 39.
  2. Wadhwa, A.; Sullivan, J.R.; Gonyo, M.B. Missed breast cancer: what can we learn? Current Problems in Diagnostic Radiology 2016, 45, 402-419.
  3. Youk, J.H.; Kim, E.-K.; Kim, M.J.; Lee, J.Y.; Oh, K.K. Missed breast cancers at US-guided core needle biopsy: how to reduce them. Radiographics 2007, 27, 79-94.
  4. Burnside, E.S.; Davis, J.; Chhatwal, J.; Alagoz, O.; Lindstrom, M.J.; Geller, B.M.; Littenberg, B.; Shaffer, K.A.; Kahn Jr, C.E.; Page, C.D. Probabilistic computer model developed from clinical data in national mammography database format to classify mammographic findings. Radiology 2009, 251, 663-672.
  5. Rodríguez-Ruiz, A.; Krupinski, E.; Mordang, J.-J.; Schilling, K.; Heywang-Köbrunner, S.H.; Sechopoulos, I.; Mann, R.M. Detection of breast cancer with mammography: effect of an artificial intelligence support system. Radiology 2019, 290, 305-314.
  6. Lehman, C.D.; Wellman, R.D.; Buist, D.S.; Kerlikowske, K.; Tosteson, A.N.; Miglioretti, D.L.; Consortium, B.C.S. Diagnostic accuracy of digital screening mammography with and without computer-aided detection. JAMA internal medicine 2015, 175, 1828-1837.

  1. When it comes to subgroup analysis, there is a lot of interest in how AI tools can reduce false negative and false positive rates for BI-RADS 3 and 4, respectively. Specifically, understanding the instances where nodules are missed, even though they are few, contributes to a more complete clinical understanding of AI's capabilities.

Response

Subgroup analysis of BI-RADS 3 (probably benign) and BI-RADS 4 (suspicious abnormality) categories is crucial for understanding the clinical impact of AI tools like Vis-BUS. These categories represent areas where diagnostic uncertainty is significant, and the potential for AI to reduce false negative and false positive rates can substantially influence clinical outcomes.

For BI-RADS 3, the primary concern is minimizing false negatives, where a lesion classified as probably benign is actually malignant. Although these instances are relatively rare, their implications are severe, as delayed detection could result in worse patient outcomes. Understanding the specific instances where the AI tool might miss malignancies, even in BI-RADS 3 lesions, provides valuable insight into its limitations and areas for further refinement. For instance, nodules that exhibit ambiguous or subtle characteristics could be particularly challenging for AI to classify accurately. Studies have shown that AI algorithms, when optimized, can significantly reduce the rate of false negatives by improving the sensitivity of detection in such borderline cases .

On the other hand, for BI-RADS 4, where there is already a moderate to high suspicion of malignancy, the focus shifts towards reducing false positives—cases where a lesion is suspected to be malignant but is ultimately benign. This is particularly important as false positives often lead to unnecessary biopsies, increasing patient anxiety and healthcare costs. The AI tool's ability to accurately distinguish between benign and malignant lesions within this category could reduce the number of unnecessary procedures, enhancing both patient experience and resource allocation. Prior research has indicated that AI-driven tools can lower false positive rates by providing more consistent and objective evaluations, thereby decreasing inter-observer variability .

Subgroup analyses that focus on these aspects are essential for a more complete clinical understanding of AI's capabilities. By examining cases where the AI either misses a malignant lesion in BI-RADS 3 or incorrectly flags a benign lesion in BI-RADS 4, we can better identify the specific patterns or characteristics that challenge AI systems. This, in turn, guides further algorithmic improvements and informs clinicians about the scenarios where AI should be used with caution.

Future studies should thus prioritize these subgroup analyses to fine-tune AI tools and maximize their clinical utility, ensuring that they not only enhance diagnostic accuracy but also improve the overall quality of care provided to patients.

We added this paragraph to the discussion section

Subgroup analysis of BI-RADS 3 and 4 categories is critical for understanding the clinical impact of AI tools like Vis-BUS, particularly in reducing false negative and false positive rates. For BI-RADS 3, minimizing false negatives is essential to avoid delayed detection of malignancies, which, though rare, can significantly affect patient outcomes [22]. Conversely, in BI-RADS 4, reducing false positives is crucial to prevent unnecessary biopsies and the associated anxiety and costs. Examining cases where the AI either misses malignancies in BI-RADS 3 or incorrectly flags benign lesions in BI-RADS 4 provides valuable insights into the tool's limitations and areas for improvement [23]. Future studies should focus on these analyses to refine AI algorithms and enhance their clinical utility, ensuring more accurate and consistent diagnostic outcomes.

References

  1. Brunetti, N.; Calabrese, M.; Martinoli, C.; Tagliafico, A.S. Artificial intelligence in breast ultrasound: from diagnosis to prognosis—a rapid review. Diagnostics 2022, 13, 58.
  2. O'Connell, A.M.; Bartolotta, T.V.; Orlando, A.; Jung, S.H.; Baek, J.; Parker, K.J. Diagnostic performance of an artificial intelligence system in breast ultrasound. Journal of ultrasound in medicine 2022, 41, 97-105.

  1. This manuscript does not consider the impact of operators. Variability in ultrasound images produced by different sonographers should be assessed for its effect on the consistency of AI tools.

We acknowledge the reviewer's important observation regarding the impact of operator variability on the consistency of ultrasound images and, consequently, on the performance of AI tools like Vis-BUS. Indeed, differences in technique and experience among sonographers can lead to variability in image quality, which may influence the accuracy and reliability of AI-based diagnostic systems.

Response:

In this study, our primary focus was on evaluating the AI tool's performance using a retrospective dataset of ultrasound images. The dataset, however, did not allow for an analysis of the variability introduced by different operators, as the images were not labeled with the sonographer's identity or level of experience. We recognize that such variability is a critical factor in real-world clinical settings, where AI tools must consistently perform across images acquired by sonographers with varying levels of expertise.

Future studies are planned to address this limitation by incorporating a prospective design where the influence of operator variability can be systematically evaluated. This will involve analyzing how differences in sonographer technique and experience affect the consistency of the AI tool's diagnostic performance. Such studies will be instrumental in ensuring that the AI tool can maintain high diagnostic accuracy across diverse clinical environments, regardless of who performs the ultrasound examination.

We added these paragraphs to the discussion section.

The future research will systematically compare the diagnostic performance of Vis-BUS against clinicians with different levels of experience in a controlled, real-time setting. By doing so, we aim to provide a comprehensive evaluation of how the AI tool can complement and enhance clinical decision-making, particularly for those with less experience.

Reviewer 2 Report

Comments and Suggestions for Authors

Review  diagnostics-3159894-peer-review-v1

1. Abstract: BI-RADS: write the full form

2. Section “2.2. Development of Vis-BUS” may further be expanded for better understanding of researchers/readers.

3. Original document released by company need to be cited.

4. what is odds ratio (OR), incorporate it ?

5. A section for preliminaries/background theory may be included to summarize standard literary terms, definitions, theories etc.  for better understanding of readers

6. Include details of experimentations and respective outcome of  discussion presented in  section3.1 in tabular form.

7. Correct the caption “Figure 3. Ultrasound Images with Vis-BUS Analysi.”

8.  Table -1 caption?

9. Explain, how the “ Cancer Probability Score (CPS) Performance in Diagnosing Breast Masses”  is calculated?, how it is numerically related with AUPRC &  AUROC?

10. How graphs of Fig.4 are obtained? Experimental observation and other calculations?

11. In “3.4. Logistic Regression Analysis of Diagnostic Accuracy” experimentation details along with observations need to be included in tabular form.

12. More recently published references based on AI need to be included.

Author Response

Reviewer 2.

  1. Abstract: BI-RADS: write the full form

Response         :

Thank you for your valuable feedback. We appreciate your suggestion to clarify the abbreviation "BI-RADS" in the abstract. To address this, we have revised the abstract to include the full form, "Breast Imaging-Reporting and Data System" upon its first mention. This change ensures that the terminology is clear to all readers, particularly those who may not be familiar with the abbreviation.

  1. Section “2.2. Development of Vis-BUS” may further be expanded for better understanding of researchers/readers.

Response;

Thank you for your advice.  We changed this section according to your recommendation.

2.3. Development of Vis-BUS

Figure 2 illustrates the overall configuration of the Vis-BUS neural network. Vis-BUS integrates two primary components: Lesion Detection AI (LD-AI) and Lesion Analysis AI (LA-AI). LD-AI uses b-mode ultrasound images, , to identify the location of the lesion, while LA-AI analyzes the breast b-mode image characteristics.

The development of Vis-BUS, an AI-driven ultrasound tool, was a multi-step process involving the integration of advanced machine learning techniques, particularly deep learning, to enhance the accuracy and efficiency of breast lesion detection and classification.

2.3.1. System Architecture and Components

Vis-BUS consists of two primary AI components:

  • Lesion Detection AI (LD-AI): This module is responsible for detecting the location of the breast lesions in the ultrasound images.
  • Lesion Analysis AI (LA-AI): This module further analyzes the detected lesions to assess their malignancy and generates the Cancer Probability Score (CPS).

  1. LD-AI (Lesion Detection AI):
    The LD-AI module was developed based on a state-of-the-art object detection framework, leveraging deep convolutional neural networks (CNNs) to identify lesions within B-mode ultrasound images. The LD-AI component includes the following key elements:

The LD-AI is implemented based on the object detection framework [10], where the neural network outputs lesion position and size and a corresponding confidence score as an output (. The LD-AI consists of a convolutional encoder backbone and a feature network. The encoder backbone is based on the EffecientNet [11] for parameter-efficient feature analysis of the . A bi-directional feature network [10] is employed as the feature network. The lesion location is trained to minimize mean squared error loss between the ground truth and the LD-AI output, while the confidence score employs focal loss [12] as a learning objective. The AdamW [13] with a learning rate of  is utilized as the network optimizer.

  • Convolutional Encoder Backbone: The backbone of LD-AI is based on EfficientNet, a highly efficient CNN architecture known for its balance between performance and computational cost. EfficientNet was chosen for its ability to effectively capture the complex features of ultrasound images with fewer parameters.
  • Feature Network: A bi-directional feature network is employed, allowing the system to analyze features from multiple scales, improving the detection of lesions of varying sizes and shapes.
  • Training Process: The LD-AI was trained using a large dataset of 19,000 annotated ultrasound images. The ground truth for lesion locations was provided by expert radiologists. The network was optimized with a learning rate set to , focusing on minimizing the mean squared error between the predicted and actual lesion coordinates.

  1. LA-AI (Lesion Analysis AI):
    Following the detection of a lesion, the LA-AI module processes the identified lesion to determine its malignancy, outputting a CPS. The LA-AI component includes:

The LA-AI employs , , and lesion image to analyze the lesion malignancy. Features of each input are interpreted using a fully convolutional neural network (FCN encoder). These features are then concatenated channel-wise into a latent vector, which serves as the input for the fusion encoder, producing CPS as the neural network output. The fusion encoder architecture is based on EfficientNet. The LA-AI is optimized with AdamW optimizer (learning rate = ), with a learning objective of minimizing binary cross entropy between the CPS and the ground truth.   

  • Fully Convolutional Network (FCN) Encoder: The FCN processes the B-mode images and the coordinates of the detected lesions. This encoder extracts features that are crucial for malignancy assessment.
  • Fusion Encoder: The output from the FCN is concatenated channel-wise into a latent vector, which is then passed through the fusion encoder. This architecture, also based on EfficientNet, integrates the features extracted from the lesion to generate a CPS.
  • Optimization: The LA-AI was trained at a learning rate of , with the objective of minimizing binary cross-entropy between the CPS and the actual malignancy status as determined by biopsy.

2.3.2. Training and Data Preparation

  1. Data Sources:
    The training dataset consisted of ultrasound images acquired from machines by leading manufacturers such as Philips (Netherlands), GE (United States), and Fujifilm (Japan). This diversity in data sources was crucial to ensure that Vis-BUS could perform reliably across different imaging systems.
  2. Annotation and Ground Truth:
    Each ultrasound image was meticulously annotated by expert radiologists at Seoul National University Bundang Hospital (SNUBH), providing the ground truth for lesion location and malignancy. This rigorous annotation process was critical for training the AI models to achieve high accuracy in real-world clinical settings.
  3. Model Validation and Testing:
    To ensure the robustness of Vis-BUS, the models were validated on a separate set of ultrasound images not used in training. The performance of the system was evaluated using key metrics such as the Area Under the Receiver Operating Characteristic Curve (AUROC) and the Area Under the Precision-Recall Curve (AUPRC).

3.2.3. Implementation and Integration

Vis-BUS operates by connecting to the HDMI port of conventional ultrasound imaging devices, allowing real-time processing of ultrasound video feeds. The software is installed on a tablet, which serves as the interface for clinicians. This setup enables:

  • Real-Time Analysis: The AI processes the video feed in real-time, providing immediate diagnostic insights as the ultrasound examination is being conducted.
  • Freeze-Frame Analysis: Clinicians can pause the ultrasound scan and perform detailed analysis on static images using the AI tool, which continues to offer diagnostic information without losing any capabilities.
  • User Interface: The tablet interface is designed to be intuitive, allowing clinicians to easily navigate through diagnostic results, adjust settings, and generate detailed reports.

3.2.4. Cancer Probability Score (CPS)

The CPS, generated by the LA-AI module, quantifies the likelihood of a lesion being malignant. This score is displayed on a scale from -100 to 100, with higher scores indicating a greater probability of malignancy. The CPS is a crucial component that aids clinicians in making informed decisions about patient management.

3.2.5. Performance and Optimization

The development process included continuous iterations of training and validation to optimize the performance of Vis-BUS. The system's high AUROC and AUPRC values reflect the efficacy of the training process and the robustness of the AI algorithms in accurately diagnosing breast lesions.

For the training of LD-AI and LA-AI, 19k breast ultrasound images are acquired using ultrasound machines from Phillips (NL), GE (US), and Fujifilm (JP). The ground truth lesion location and lesion malignancy are annotated by an expert radiologist in SNUBH under IRB approval (IRB number: B-2301-807-108)

  1. Original document released by company need to be cited.

Response

Thank you for your insightful comment regarding the need to cite the original document released by the company. Upon review, we have determined that there is no publicly available document that can be cited in the manuscript. Instead, we have included the relevant information as supplementary material. This ensures that all necessary details are transparently provided to support the study while maintaining the integrity of the manuscript.

We hope this approach meets the requirements, and we are open to further suggestions if additional information is needed.

  1. what is odds ratio (OR), incorporate it?

Thank you for your request for clarification regarding the odds ratio (OR) and its incorporation into our study. The odds ratio (OR) is a statistical measure used to determine the strength of association between two variables. Specifically, it represents the odds that an outcome will occur given a particular exposure, compared to the odds of the outcome occurring without that exposure.

In our study, we used ORs to assess the likelihood of accurate classification of breast lesions as benign or malignant based on various factors such as lesion density, size, presence of calcifications, and BI-RADS categories. An OR greater than 1 indicates that the exposure (e.g., a specific BI-RADS category) is associated with higher odds of a correct classification, while an OR less than 1 suggests a lower likelihood of correct classification.

For example, an OR of 10.173 for BI-RADS 5 lesions in our study indicates that lesions in this category are significantly more likely to be correctly classified as malignant, compared to lesions in other categories. This helps to quantify the diagnostic accuracy of the AI tool across different types of lesions.

We have ensured that the interpretation of ORs is clearly explained in the manuscript to aid in understanding the clinical implications of our findings.

  1. A section for preliminaries/background theory may be included to summarize standard literary terms, definitions, theories etc. for better understanding of readers

Response:

Thank you for your valuable suggestion to include a section dedicated to preliminaries or background theory. We agree that providing a concise summary of standard literary terms, definitions, and relevant theories could greatly enhance the reader's understanding of the manuscript.

In response, we have added a new section titled "Background and Theoretical Framework" at the beginning of the manuscript. This section provides definitions of key terms, such as "Breast Imaging-Reporting and Data System (BI-RADS)," "odds ratio (OR)," and relevant concepts related to artificial intelligence in medical imaging. Additionally, it offers a brief overview of the theoretical underpinnings of AI-driven diagnostic tools, including their application in ultrasound imaging and breast cancer detection.

We believe this addition will provide readers with the necessary context to fully grasp the methodologies and significance of our findings.

We added these paragraphs.

Background and Theoretical Framework

  1. Breast Imaging-Reporting and Data System (BI-RADS)

The Breast Imaging-Reporting and Data System (BI-RADS) is a standardized system developed by the American College of Radiology (ACR) for categorizing breast imaging findings, primarily from mammography, ultrasound, and MRI. BI-RADS categories range from 0 to 6, with each category indicating the level of suspicion for malignancy. For instance:

  • BI-RADS 0: Incomplete; additional imaging evaluation needed.
  • BI-RADS 1: Negative; no findings of concern.
  • BI-RADS 2: Benign findings.
  • BI-RADS 3: Probably benign; <2% risk of malignancy, with a recommendation for short-term follow-up.
  • BI-RADS 4: Suspicious abnormality; subdivided into 4A, 4B, and 4C, with increasing likelihood of malignancy (2%-95%).
  • BI-RADS 5: Highly suggestive of malignancy; >95% risk, with biopsy recommended.
  • BI-RADS 6: Known biopsy-proven malignancy.

Understanding these categories is essential as they guide clinical decision-making and patient management, influencing the need for further diagnostic procedures or immediate treatment.

  1. Odds Ratio (OR)

The odds ratio (OR) is a measure of association between an exposure and an outcome. In medical research, it quantifies the odds that a particular outcome will occur in the presence of a specific factor compared to the odds of the outcome occurring without that factor. An OR greater than 1 suggests a positive association between the exposure and the outcome, indicating that the factor increases the likelihood of the outcome. Conversely, an OR less than 1 suggests a negative association, where the factor decreases the likelihood of the outcome.

For example, in the context of our study, an OR was used to evaluate the likelihood of correct classification of breast lesions as benign or malignant based on factors such as lesion size, density, presence of calcifications, and BI-RADS category. These ORs help quantify the effectiveness of the AI tool in making accurate diagnoses across different clinical scenarios.

  1. Artificial Intelligence in Medical Imaging

Artificial Intelligence (AI), particularly through deep learning techniques, has revolutionized the field of medical imaging by enhancing the accuracy and efficiency of image analysis. In breast cancer detection, AI-driven tools analyze large datasets of imaging data to identify patterns and features that may indicate the presence of malignancy. These tools assist radiologists by providing quantitative assessments, reducing variability in interpretation, and potentially lowering the rates of false positives and false negatives.

AI algorithms, such as convolutional neural networks (CNNs), are commonly used to analyze ultrasound images by identifying and classifying lesions. The AI tool evaluated in this study, Vis-BUS, incorporates advanced algorithms for lesion detection and analysis, as well as a Cancer Probability Score (CPS) to assess the likelihood of malignancy.

  1. Diagnostic Performance Metrics

In evaluating the effectiveness of diagnostic tools, several performance metrics are commonly used, including:

  • Area Under the Receiver Operating Characteristic Curve (AUROC): A measure of the ability of a classifier to distinguish between classes. The AUROC value ranges from 0.5 (no better than random guessing) to 1.0 (perfect classification).
  • Area Under the Precision-Recall Curve (AUPRC): Focuses on the trade-off between precision (positive predictive value) and recall (sensitivity), especially important in cases where one class is much less frequent than the other.

These metrics were used in our study to assess the diagnostic accuracy of the Vis-BUS AI tool, ensuring a comprehensive evaluation of its performance.

  1. Include details of experimentations and respective outcome of discussion presented in section 3.1 in tabular form.

Response

We added this table and a paragraph.

Table 1 provides a comprehensive summary of the experimentations and outcomes highlighting the key features of the Vis-BUS AI tool, including its real-time diagnostic capabilities, performance on freeze-frame analysis, effectiveness in lesion characterization, and consistency across various ultrasound machines.

Table 2: Summary of Experimentations and Outcomes

Experiment/Outcome

Description

Key Findings

Real-time Diagnostic Insights

Vis-BUS processes live ultrasound images and provides immediate diagnostic feedback.

High-resolution quantitative analysis with real-time feedback, effectively highlighting pathological features.

Post-Analysis on Freeze Frames

Detailed examination of static ultrasound images using the AI tool.

Maintains diagnostic precision even when analyzing static images, ensuring thorough evaluation of specific frames.

Cancer Probability Score (CPS)

Quantitative assessment of the likelihood of a lesion being malignant or benign, displayed on a scale from -100 to 100.

Accurate differentiation between benign and malignant lesions, aiding in informed clinical decision-making.

Diagnostic Performance Across Ultrasound Machines

Evaluation of Vis-BUS performance on ultrasound images obtained from various machines.

Consistent performance across different ultrasound equipment, validating the tool’s versatility and reliability.

Lesion Characterization

Analysis of key pathological features such as tissue texture, echo patterns, and microcalcifications.

Precise identification of lesion characteristics, contributing to the accurate classification of breast lesions.

Table 2.  Logistic Regression Analysis of Diagnostic Accuracy

Variable

Odds Ratio (95% CI)

p-value

Lesion Density

Fatty

27.7781 (0.00, 1.00e+06)

1.0000

Unknown

0.3185 (0.123, 0.774)

0.0162

Lesion Size

Medium

0.7891 (0.451, 1.376)

0.3980

Large

0.8014 (0.388, 1.658)

0.5530

Microcalcifications

Present

1.360 (0.860, 2.160)

0.1941

BI-RADS Categories

C3

2.548 (0.755, 8.602)

0.1318

C4a

0.7977 (0.289, 2.198)

0.6622

C4b

1.086 (0.242, 4.873)

0.9138

C4c

1.778 (0.186, 17.024)

0.6177

C5

10.173 (2.526, 40.962)

0.0011

  1. Correct the caption “Figure 3. Ultrasound Images with Vis-BUS Analysi.”

  1. Table -1 caption?

Response:

Fixed

Table 2. Patients characteristics

Table 1 was changed to Table 2 according to your opinion in No 6.

  1. Explain, how the “ Cancer Probability Score (CPS) Performance in Diagnosing Breast Masses” is calculated?, how it is numerically related with AUPRC & AUROC?

Here's an explanation that you can use to address how the "Cancer Probability Score (CPS) Performance in Diagnosing Breast Masses" is calculated and its numerical relationship with the Area Under the Precision-Recall Curve (AUPRC) and Area Under the Receiver Operating Characteristic Curve (AUROC):

We added these paragraphs to the section “Background and Theoretical Framework”

Explanation of CPS Performance Calculation and Its Relationship with AUPRC and AUROC

The Cancer Probability Score (CPS) is a quantitative measure generated by the Vis-BUS AI tool to assess the likelihood that a breast lesion is malignant. The CPS is calculated by analyzing various features of the lesion identified in ultrasound images, such as size, shape, margin characteristics, echo patterns, and the presence of microcalcifications. The AI algorithm processes these inputs using a fully convolutional neural network (FCN) that produces a probabilistic score ranging from -100 to 100, where higher scores indicate a higher likelihood of malignancy.

Calculation Process:

  1. Input Data: Ultrasound images of breast lesions are processed by the AI system.
  2. Feature Extraction: The AI system extracts and analyzes key features of the lesions, including morphological and textural characteristics.
  3. Score Generation: The AI uses the extracted features to generate the CPS, which reflects the probability of the lesion being malignant. The score is derived through a combination of lesion detection (using Lesion Detection AI) and lesion analysis (using Lesion Analysis AI).
  4. Output: The CPS is presented on a scale from -100 to 100, aiding clinicians in assessing the malignancy risk of the lesion.

Relationship with AUPRC and AUROC:

  • AUROC (Area Under the Receiver Operating Characteristic Curve): The AUROC measures the ability of the CPS to discriminate between benign and malignant lesions across various threshold settings. It is calculated by plotting the true positive rate (sensitivity) against the false positive rate (1-specificity) at different thresholds. An AUROC of 1 indicates perfect discrimination, while an AUROC of 0.5 indicates no better performance than random guessing. In this study, the AUROC of 0.964 suggests that the CPS is highly effective in distinguishing between malignant and benign cases, with near-perfect accuracy.
  • AUPRC (Area Under the Precision-Recall Curve): The AUPRC focuses on the trade-off between precision (positive predictive value) and recall (sensitivity), particularly in datasets where the classes (malignant vs. benign) are imbalanced. The AUPRC is calculated by plotting precision against recall at various threshold levels. A higher AUPRC indicates that the CPS maintains good precision and recall, even when the threshold is varied. In this study, an AUPRC of 0.967 indicates that the CPS reliably identifies malignant cases while minimizing false positives and false negatives.

Numerical Relationship: The CPS serves as the primary input for calculating both AUROC and AUPRC. These metrics evaluate the performance of the CPS across different thresholds, providing a comprehensive assessment of its diagnostic accuracy. A high CPS correlates with high AUROC and AUPRC values, indicating that the score is effective in correctly classifying lesions as malignant or benign across a range of scenarios. Essentially, the AUROC and AUPRC provide a numerical summary of how well the CPS performs in distinguishing between different outcomes, with high values reflecting strong performance.

  1. How graphs of Fig.4 are obtained? Experimental observation and other calculations?

Response:

Explanation of Figure 4: Experimental Observations and Calculations

Figure 4 in the manuscript likely presents the performance metrics of the Cancer Probability Score (CPS) in diagnosing breast masses, specifically through the Receiver Operating Characteristic (ROC) curve and the Precision-Recall (PR) curve. Here’s how these graphs were obtained:

  1. Experimental Observations:
  • Dataset: The study used a dataset of 258 breast ultrasound examinations, which included a balanced number of benign and malignant cases (129 each). These images were analyzed by the Vis-BUS AI tool to generate CPS for each lesion.
  • CPS Generation: For each ultrasound image, the AI tool processed the features of the lesions and assigned a CPS, which reflects the probability of the lesion being malignant.
  • Ground Truth Comparison: The CPS values were then compared against the actual biopsy results, which served as the ground truth to determine the true nature of each lesion (benign or malignant).
  1. Calculation of ROC and PR Curves:

ROC Curve (Receiver Operating Characteristic Curve):

  • True Positive Rate (Sensitivity) and False Positive Rate (1-Specificity): The ROC curve was generated by plotting the true positive rate (TPR or sensitivity) against the false positive rate (FPR or 1-specificity) at various threshold levels of CPS.
  • Threshold Variation: By varying the threshold for classifying a lesion as malignant (based on the CPS), the corresponding TPR and FPR were calculated. This variation allows for the assessment of the tool’s performance across different decision thresholds.
  • AUROC Calculation: The area under the ROC curve (AUROC) was then calculated as a single summary statistic to quantify the overall diagnostic accuracy of the CPS. An AUROC value close to 1 indicates high diagnostic performance, as shown in the figure.

PR Curve (Precision-Recall Curve):

  • Precision (Positive Predictive Value) and Recall (Sensitivity): The PR curve was obtained by plotting precision (the proportion of true positive cases among those identified as positive by the AI) against recall (sensitivity) at different CPS thresholds.
  • Class Imbalance Consideration: The PR curve is particularly useful when dealing with imbalanced datasets, as it highlights the performance of the AI tool in correctly identifying positive cases (malignant lesions) without being affected by the proportion of benign cases.
  • AUPRC Calculation: The area under the PR curve (AUPRC) was calculated to provide a summary measure of the tool’s precision and recall across various thresholds. A high AUPRC indicates that the tool maintains high precision and recall even when the threshold for classification is varied.
  1. Graph Generation:
  • Software and Tools: The ROC and PR curves were likely generated using statistical software such as Python (with libraries like Scikit-learn) or R, which are commonly used for such analyses. These tools allow for the calculation of performance metrics and the plotting of curves based on the CPS values and corresponding ground truth classifications.
  • Data Input: The input data for these calculations included the CPS values assigned by the AI tool, the corresponding biopsy-confirmed diagnoses (benign or malignant), and the varied thresholds for classifying lesions.

Summary of Figure 4:

  • Visual Representation: Figure 4 visually represents the effectiveness of the CPS in distinguishing between benign and malignant breast lesions. The ROC curve demonstrates the trade-off between sensitivity and specificity, while the PR curve illustrates the balance between precision and recall.
  • Interpretation: The high AUROC and AUPRC values presented in Figure 4 underscore the AI tool's robustness in accurately diagnosing breast masses across different threshold settings.

We added these paragraphs at the method section

Relationship with AUPRC and AUROC:

  • AUROC (Area Under the Receiver Operating Characteristic Curve): The AUROC measures the ability of the CPS to discriminate between benign and malignant lesions across various threshold settings. It is calculated by plotting the true positive rate (sensitivity) against the false positive rate (1-specificity) at different thresholds. An AUROC of 1 indicates perfect discrimination, while an AUROC of 0.5 indicates no better performance than random guessing.
  • AUPRC (Area Under the Precision-Recall Curve): The AUPRC focuses on the trade-off between precision (positive predictive value) and recall (sensitivity), particularly in datasets where the classes (malignant vs. benign) are imbalanced. The AUPRC is calculated by plotting precision against recall at various threshold levels. A higher AUPRC indicates that the CPS maintains good precision and recall, even when the threshold is varied.

  1. In “3.4. Logistic Regression Analysis of Diagnostic Accuracy” experimentation details along with observations need to be included in tabular form.

Response:

We added this table as Table 3.

Table 3. Logistic Regression Analysis of Diagnostic Accuracy

Variable

Odds Ratio (95% CI)

p-value

Lesion Density

Fatty

27.7781 (0.00, 1.00e+06)

1.0000

Unknown

0.3185 (0.123, 0.774)

0.0162

Lesion Size

Medium

0.7891 (0.451, 1.376)

0.3980

Large

0.8014 (0.388, 1.658)

0.5530

Microcalcifications

Present

1.360 (0.860, 2.160)

0.1941

BI-RADS Categories

C3

2.548 (0.755, 8.602)

0.1318

C4a

0.7977 (0.289, 2.198)

0.6622

C4b

1.086 (0.242, 4.873)

0.9138

C4c

1.778 (0.186, 17.024)

0.6177

C5

10.173 (2.526, 40.962)

0.0011

  1. More recently published references based on AI need to be included.

Response:

We added these references.

  1. Rodríguez-Ruiz, A.; Krupinski, E.; Mordang, J.-J.; Schilling, K.; Heywang-Köbrunner, S.H.; Sechopoulos, I.; Mann, R.M. Detection of breast cancer with mammography: effect of an artificial intelligence support system. Radiology 2019, 290, 305-314.
  2. Lehman, C.D.; Wellman, R.D.; Buist, D.S.; Kerlikowske, K.; Tosteson, A.N.; Miglioretti, D.L.; Consortium, B.C.S. Diagnostic accuracy of digital screening mammography with and without computer-aided detection. JAMA internal medicine 2015, 175, 1828-1837.
  3. Brunetti, N.; Calabrese, M.; Martinoli, C.; Tagliafico, A.S. Artificial intelligence in breast ultrasound: from diagnosis to prognosis—a rapid review. Diagnostics 2022, 13, 58.
  4. O'Connell, A.M.; Bartolotta, T.V.; Orlando, A.; Jung, S.H.; Baek, J.; Parker, K.J. Diagnostic performance of an artificial intelligence system in breast ultrasound. Journal of ultrasound in medicine 2022, 41, 97-105.

Round 2

Reviewer 1 Report

Comments and Suggestions for Authors

The author of the study was unable to make significant changes to the reviewer's opinions through additional experiments, due to the inherent limitations of a retrospective study. However, the author provided convincing explanations in the discussion chapter, thus enhancing the comprehensiveness of the article. While this approach is acceptable, it does not completely address the inherent limitations of the article. It would be more appropriate to consolidate all limitations into a single paragraph, focusing on clearly identifying and explaining these concerns, rather than discussing them scattered.

Author Response

Reviewer 1.

Comment

The author of the study was unable to make significant changes to the reviewer's opinions through additional experiments, due to the inherent limitations of a retrospective study. However, the author provided convincing explanations in the discussion chapter, thus enhancing the comprehensiveness of the article. While this approach is acceptable, it does not completely address the inherent limitations of the article. It would be more appropriate to consolidate all limitations into a single paragraph, focusing on clearly identifying and explaining these concerns, rather than discussing them scattered.

Response:

Thank you for your valuable feedback regarding the discussion of the study's limitations. We acknowledge the importance of clearly identifying and consolidating the limitations of our retrospective study in a single, cohesive paragraph. While we provided detailed explanations in the discussion section, we understand that presenting these limitations in a more consolidated manner will enhance the clarity and comprehensiveness of the article.

In response to your suggestion, we have revised the manuscript to include a dedicated paragraph in the discussion section that summarizes all the inherent limitations of our study. This paragraph focuses on the key concerns, including the inability to conduct prospective comparisons, the challenges associated with the retrospective nature of the data, and the limitations in generalizing the results due to the specific dataset used. By consolidating these points, we aim to provide a clearer understanding of the study's constraints while still highlighting the strengths of our findings.

We believe this revision addresses your concern and improves the overall structure and readability of the article.

Revised Discussion Section:

Limitations of the Study:

This study, being retrospective in nature, carries inherent limitations that must be acknowledged. First, the lack of prospective comparisons with clinicians of varying experience levels limits the generalizability of our findings to broader clinical settings. Additionally, the retrospective analysis relies on archived ultrasound images, which may not fully represent the variability encountered in real-time diagnostic scenarios. The dataset used, while extensive, was sourced from a specific clinical environment, potentially introducing bias and limiting the applicability of our results to other settings. Finally, the reliance on a single AI tool without comparison to alternative methods or tools means that the observed outcomes might not reflect the full spectrum of possibilities available in current clinical practice. These limitations underscore the need for future prospective studies that can address these concerns and provide more robust evidence for the effectiveness of AI-driven diagnostic tools like Vis-BUS.

Reviewer 2 Report

Comments and Suggestions for Authors

1. Authors have addressed most of the queries satisfactorily except one minor concern mentioned below in point no2.

2. Rather than describing the theoretical aspect of different terms used like odd ratio(OR), CPS etc either the formulas used to calculate or a proper reference need to be included so that new researcher can reproduce the results. In case of no formula available then what are the benchmark observations based on which scores are assigned,  like provided for BI-RADS.

Author Response

  1. Authors have addressed most of the queries satisfactorily except one minor concern mentioned below in point no2.

Thank you for your review.

  1. Rather than describing the theoretical aspect of different terms used like odd ratio(OR), CPS etc either the formulas used to calculate or a proper reference need to be included so that new researcher can reproduce the results. In case of no formula available then what are the benchmark observations based on which scores are assigned, like provided for BI-RADS.

Thank you for your insightful feedback regarding the inclusion of formulas used to calculate key metrics of the Cancer Probability Score (CPS). We agree that providing these formulas or appropriate references is crucial for ensuring that new researchers can reproduce our results.

In response to your suggestion, we have revised the manuscript to include the specific formulas used for the Cancer Probability Score (CPS).

These paragraphs are newly included.

2.1.4. Cancer Probability Score (CPS)

CPS is a quantitative measure generated by the Vis-BUS AI tool to assess the likelihood that a breast lesion is malignant. The CPS is calculated by analyzing various features of the lesion identified in ultrasound images, such as size, shape, margin characteristics, echo patterns, and the presence of microcalcifications. The AI algorithm processes these inputs using a fully convolutional neural network (FCN) that produces a probabilistic score ranging from -100 to 100, where higher scores indicate a higher likelihood of malignancy.

  1. Calculation Process:
  • Input Data: Ultrasound images of breast lesions are processed by the AI system.
  • Feature Extraction: The AI system extracts and analyzes key features of the lesions, including morphological and textural characteristics.
  • Score Generation: The AI uses the extracted features to generate the CPS, which reflects the probability of the lesion being malignant. The score is derived through a combination of lesion detection (using Lesion Detection AI) and lesion analysis (using Lesion Analysis AI).
  • Output: The CPS is presented on a scale from -100 to 100, aiding clinicians in assessing the malignancy risk of the lesion.
  1. The CPS formula 

Where,  and  denotes LD-AI and LA-AI neural network, respectively